# Variational Last Layers for Bayesian Optimization

**Paul Brunzema**
RWTH Aachen University
brunzema@dsme.rwth-aachen.de

**Mikkel Jordahn**
Technical University of Denmark
jordahnm@gmail.com

**John Willes**
Vector Institute for AI
john.willes@vectorinstitute.ai

**Sebastian Trimpe**
RWTH Aachen University
trimpe@dsme.rwth-aachen.de

**Jasper Snoek**
Google DeepMind
jsnoek@google.com

**James Harrison**
Google DeepMind
jamesharrison@google.com

## Abstract

Gaussian Processes (GPs) are widely seen as the state-of-the-art surrogate models for Bayesian optimization (BO) due to their ability to model uncertainty and their performance on tasks where correlations are easily captured, such as those defined by Euclidean metrics. However, the performance of GPs depends on the choice of kernel, and kernel selection for complex correlation structures is often difficult. While Bayesian neural networks are a promising direction for higher capacity surrogate models, they have so far seen limited use due to a combination of cost of use and poor performance. In this paper, we explore the potential of neural networks with variational Bayesian last layers (VBLLs), which offer a simple and computationally lightweight approach to Bayesian uncertainty quantification in neural networks. Our findings suggest that VBLL networks significantly outperform GPs and other BNN architectures on tasks with complicated input correlations, and match the performance of well-tuned GPs on established benchmark tasks. These results highlight their promise as an alternative surrogate model for BO.

## 1 Introduction

Bayesian optimization (BO) has become an immensely popular method for optimizing black-box functions that are expensive to evaluate, and has seen large success in a variety of applications [1–10]. In BO, the goal is to optimize some black-box objective $f\colon \mathcal{X} \to \mathbb{R}$ (where $\mathcal{X} \subseteq \mathbb{R}^d$) in as few samples as possible whilst only having access to sequentially sampled, potentially noisy, data points from the objective. Gaussian processes (GPs) have long been the de-facto surrogate models in BO due to their well-calibrated uncertainty quantification and strong performance in small-data regimes. However, their application becomes challenging in high-dimensional, non-stationary, and structured data environments such as drug-discovery [10, 9] and materials science [5]. Here, often prohibitively expensive or bespoke kernels are necessary to capture meaningful correlations between data points. Furthermore, the scaling of GPs to large datasets typically associated with high-dimensional spaces can be limiting–especially if combined with online hyperparameter estimation. To address these challenges, integrating Bayesian Neural Networks (BNNs) into BO as alternative surrogate models has gained increasing attention [11–14]. While BNNs inherently scale with data, challenges like efficiently conditioning on new data and consistency across tasks persist. Our work demonstrates how

Workshop on Bayesian Decision-making and Uncertainty, 38th Conference on Neural Information Processing Systems (NeurIPS 2024).

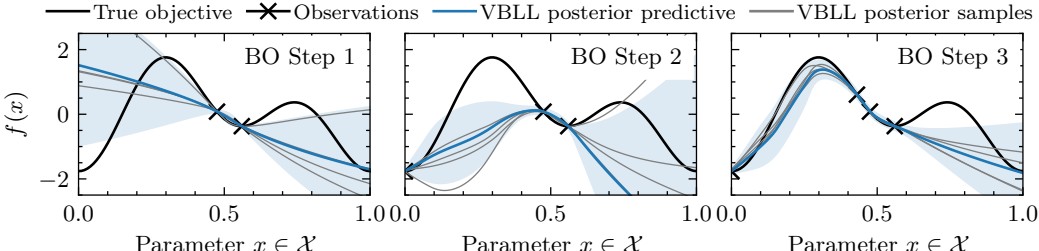

Figure 1: Variational Bayesian last layer model as a surrogate model for BO on a toy example. The VBLL model can capture *in-between* uncertainty and analytic posterior samples are easily obtained through its parametric form making it a suitable surrogate for BO.

Variational Bayesian Last Layer (VBLL) neural networks [15] can address these issues and achieve state-of-the-art performance with the same architecture across various optimization problems.

**Contributions:** In this work we investigate the use of VBLL neural networks [15] for the first time in BO (cf. Fig. 1), and explore avenues for further improving these models. In particular, our main contributions and findings are: *(i)* VBLL models outperform I-BNN models, a recently proposed surrogate model [14], on smooth synthetic benchmarks; *(ii)* VBLL models outperform GPs on problems with complex input correlations; *(iii)* VBLL models (and NN surrogate models generally) are sensitive to the training strategy used. Finally, we discuss implications for future work.

## 1.1 Related Work and Background

Various flavors of Bayesian or partially-Bayesian networks have been explored for BO, including mean field BNNs [13], networks trained via stochastic gradient Hamiltonian Monte Carlo [12, 16], and last layer Laplace approximation BNNs [17, 18]. In [14], the authors find that infinite-width BNNs (I-BNNs) [19–21], perform particularly well especially on high-dimensional, non-stationary and non-Euclidean problems, a setting where standard GPs tend to struggle.

While BNNs are promising, they have often proven to be challenging to train and complex to use in practice. Bayesian last layer networks—which consider uncertainty only over the output layer—provide a simple (and often much easier to train) partially-Bayesian neural network model [11, 22–26]. Concretely, the standard model for regression with Bayesian last layer networks and a one-dimensional output is $y = \boldsymbol{w}^\top \boldsymbol{\phi_\theta}(\boldsymbol{x}) + \varepsilon$, where $\boldsymbol{w} \in \mathbb{R}^m$, and $\boldsymbol{\phi_\theta}$ are the features learned by a neural network backbone with parameters $\boldsymbol{\theta}$. The noise $\varepsilon \sim \mathcal{N}(\boldsymbol{0}, \Sigma)$ is assumed to be independent and identically distributed. With this observation model, fixed features $\boldsymbol{\phi_\theta}$, and a Gaussian prior on the weights as $p(\boldsymbol{w}) = \mathcal{N}(\bar{\boldsymbol{w}}, S)$, posterior inference for the weights is analytically tractable via Bayesian linear regression, yielding the posterior $p(\boldsymbol{w} \mid X, Y) = \mathcal{N}(\bar{\boldsymbol{w}}, S)$ and posterior predictive

$$p(y \mid \boldsymbol{x}, \boldsymbol{\eta}, \boldsymbol{\theta}) = \mathcal{N}(\bar{\boldsymbol{w}}^\top \boldsymbol{\phi_\theta}(\boldsymbol{x}),\ \boldsymbol{\phi_\theta}(\boldsymbol{x})^\top S \boldsymbol{\phi_\theta}(\boldsymbol{x}) + \Sigma) \quad \text{where} \quad \boldsymbol{\eta} := (\bar{\boldsymbol{w}}, S). \tag{1}$$

Since the predictive distribution is Gaussian, it pairs nicely with conventional acquisition functions in Bayesian optimization and bandit tasks [11, 25].

Usually, such BLL models are trained using gradient descent on the exact (log) marginal likelihood over all data points either via $\nabla_{\boldsymbol{\theta}} \log p(Y \mid X, \boldsymbol{\theta})$ [23] (which is computationally expensive and often unstable) or mini-batches [11] (which yields biased gradients and results in over-concentration when the model is conditioned on all data). To increase efficiency, recent work [15, 27] developed a deterministic variational lower bound to the exact marginal likelihood and proposed to optimize this instead resulting in the *variational* Bayesian last layer (VBLL) model. Following [15, Theorem 1], the variational lower bound for regression with BLLs (under the prior defined previously) is

$$\log p(Y \mid X, \boldsymbol{\theta}) \geq \sum_t^T \left( \log \mathcal{N}(y_t \mid \bar{\boldsymbol{w}}^\top \boldsymbol{\phi}_t, \Sigma) - \frac{1}{2} \boldsymbol{\phi}_t^\top S \boldsymbol{\phi}_t \Sigma^{-1} \right) - \mathrm{KL}(q(\boldsymbol{w} \mid \boldsymbol{\eta}) \parallel p(\boldsymbol{w})) \tag{2}$$

where $\boldsymbol{\phi}_t := \boldsymbol{\phi_\theta}(\boldsymbol{x}_t)$ and $q(\boldsymbol{w} \mid \boldsymbol{\eta}) = \mathcal{N}(\bar{\boldsymbol{w}}, S)$ is the variational posterior. The variational posterior over the last layer is trained with the network weights $\boldsymbol{\theta}$ yielding a lightweight Bayesian formulation.

## 2 VBLLs for Bayesian Optimization

In this section we discuss necessary algorithmic design considerations in using VBLLs within BO. In particular, we discuss network training (including accelerating training via continual learning), choice and computation of acquisition function, and model hyperparameters. We expand on all design decisions in the Appendix.

**Training:** Training of the BNNs usually follows standard neural network training, and uses a held-out validation set to determine when to stop [27, 15]. In the low data regime of BO, such an approach is not feasible as the training data size is prohibitively low, especially at the start of optimization. We therefore employ early stopping for the VBLLs based on the training loss; details of training are provided in Appendix B. The naive approach to training requires training a new network for each new data point; we therefore further explore using continual learning for faster convergence. For this, we initialize the VBLLs at iteration $k + 1$ with the variational posterior and weights of the backbone from the previous iteration $k$ as a warm start to the optimization.

**Acquisition Functions:** VBLLs have a Gaussian predictive distribution and thus most acquisition functions that are straightforward to compute for GPs are also straightforward for VBLLs. However, parametric VBLLs are also especially well suited for Thompson sampling compared to non-parametric models like GPs[1]. For Thompson sampling, we simply sample from the variational posterior of $\boldsymbol{w}$ at iteration $k$ and then construct a sample from the predictive $\hat{f}_i$ (cf. Fig. 1) as a generalized linear model as

$$\text{①} \quad \hat{\boldsymbol{w}}_i \sim q_k(\boldsymbol{w} \mid \boldsymbol{\eta}) \qquad \text{②} \quad \hat{f}_i(\boldsymbol{x}) := \hat{\boldsymbol{w}}_i^\top \boldsymbol{\phi}_{\boldsymbol{\theta}}(\boldsymbol{x}) \qquad (3)$$

This sample of the predictive can then be optimized *directly* using first-order methods, which differs from classic Thompson sampling methods used for non-parametric GPs. For the analytic optimization of the parametric sample, we use L-BFGS-B [31] leveraging the fact that we can also easily obtain the gradient of $\hat{f}$. We initialize the optimization at multiple random initial conditions[2] and choose the best argmax as the next query location.

**Hyperparameters:** The VBLL models have several hyperparameters, some of which are similar to GPs and some of which are substantially different. We include hyperparameter studies in Appendix E. These cover the parameters of the noise covariance prior, which has a reasonably substantial impact on the (point) noise covariance estimate, and comparisons of different neural network sizes.

## 3 Experiments

We evaluate the performance of the VBLL surrogate model on various standard benchmarks and three more complex optimization problems, where the optimization landscape is non-stationary. For experimental details, ablations of hyperparameters, and further results (including other acquisition functions), we refer to Appendix D and E. The baselines for all benchmarks are the following:

**GPs:** As the de-facto standard in BO, we compare against GPs. As kernel, we choose a Matérn kernel with $\nu = 2.5$ and use individual lengthscales for all input dimensions that are optimized within box constraints following recommended best practices [28, 32] (cf. Appendix A). We expect the performance of GPs to be particularly good on stationary benchmarks.

**I-BNNs:** We compare against infinite-width Bayesian neural networks (I-BNNs) [33], which have shown promising results in recent work [14]. As in Li et al. [14], we set the depth to 3 and initialize the weight variance to 10 and the bias variance to 1.6. Note that this model is still non-parametric.

**VBLL:** For the VBLLs, we use 3 layers with 128 neurons and ELU activations for all experiments to closely match the architectures of the baselines used in [14]. We compare two baselines: relearning

---

[1]Thompson sampling for GPs often involves drawing samples from high-dimensional posterior distributions generated at pseudo-random input locations (e.g., using Sobol sequences) and then selecting the argmax of the discrete samples as the next query locations [28]. It is worth noting that while it is possible to construct analytic approximate posterior samples for GPs [29, 30], this approach is not yet commonly adopted in current practice.

[2]In the following, the number of random initial conditions will be set to the same number of random initial conditions as for the standard optimization of the acquisition functions for a fair comparison.

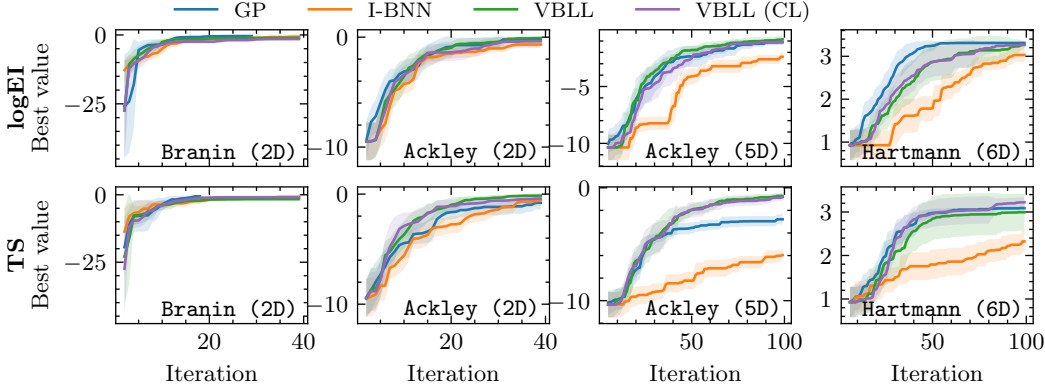

Figure 2: *Classic benchmarks.* Performance of all surrogates for `logEI` (top) and `TS` (bottom).

both features and the variational posterior from scratch at each iteration, and relearning every 5 iterations and using continual learning (CL) in between.

In all subsequent experiments, we set the number of initial points equal to the input dimensionality and the batch size to one. We compare the performance of all surrogates for the following acquisition functions: *(i)* log expected improvement (`logEI`) [34], a numerically more stable version of standard expected improvement, *(ii)* upper confidence bound (`UCB`) [35] with constant $\beta = 2$, *(iii)* and Thompson sampling (`TS`) [36, 37]. The results for `UCB` are in Appendix D.

### 3.1 Problem Settings

**Benchmark Problems:** We begin by examining a set of standard benchmark problems commonly used to assess the performance of BO algorithms [28, 34]. Figure 2 illustrates the performance of all surrogates on these benchmark problems. It can be observed that, as expected, GPs perform well. The BNN baselines also demonstrate strong performance on lower-dimensional problems, although they do not match the performance of GPs on the `Hartmann` function. Interestingly, for `TS`, we notice that on the `Ackley5D` benchmark, the VBLLs with analytic optimization of the Thompson samples even surpass the performance of GPs. The continual learning baseline shows the same performance as the standard VBLLs but with reduced compute.

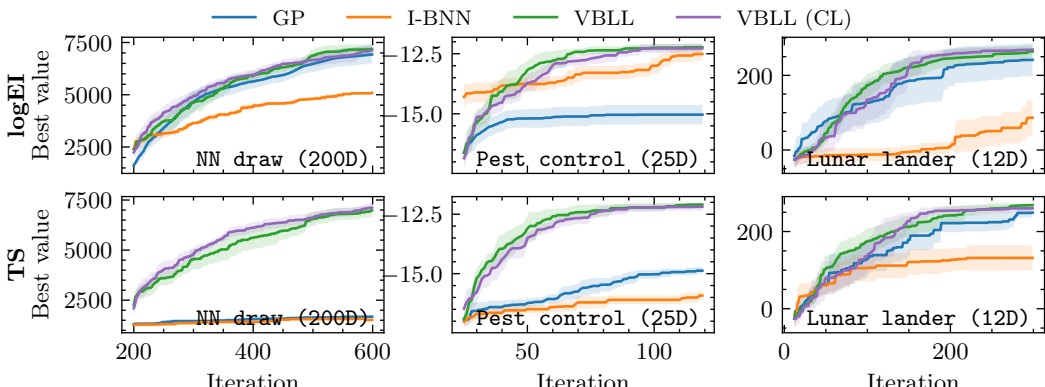

Figure 3: *High-dimensional and non-stationary benchmarks.* Performance of all surrogates for `logEI` (top) and `TS` (bottom). VBLLs demonstrate strong performance on high-dimensional problems.

**High-Dimensional and Non-Stationary Problems:** GPs without tailored kernels often struggle in high-dimensional and non-stationary environments [14]; areas where deep learning approaches are expected to excel. Our results on the 200D `NNdraw` benchmark [14], the real-world 25D `Pestcontrol` benchmark [38], and the 12D `Lunarlander` benchmark [28] are shown in Fig. 3. On these bench-

marks, VBLLs significantly outperform the other baselines; especially for `TS`. While GPs perform well with `logEI` on `NNdraw` and the I-BNNs show good performance on `Pestcontrol`, the VBLLs are consistently the best performing surrogate. Similar to the classic benchmarks, the continual learning version of the VBLLs shows similar performance to the VBLLs.

## 4  Discussion

In this paper, we explored using the recently proposed VBLLs for BO. Our findings show that VBLLs perform on par with GPs on standard low-dimensional benchmarks, yet significantly outperform GPs in high-dimensional and non-stationary problems. Furthermore, VBLLs outperform I-BNNs, a recently proposed BNN surrogate. In future work, we will explore concepts such as variational continual learning [39] to reduce the computational time for VBLLs. Finding the optimal balance between reinitializing the network and applying continual learning updates–whether through recursive updates of the last layer or re-learning the variational posterior–will be crucial for effectively integrating VBLLs into BO for real-world problems.

## Acknowledgments

The authors thank F. Solowjow, A. von Rohr, and Zi Wang for their helpful comments and discussions. P. Brunzema is partially funded by the Deutsche Forschungsgemeinschaft (DFG, German Research Foundation)–RTG 2236/2 (UnRAVeL). Simulations were performed in part with computing resources granted by RWTH Aachen University under project rwth1579.

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

# A  Further Details on the Baselines and Acquisition Functions

We implement all baselines and experiment in BoTorch [32] and GPyTorch [40]. As best-practice in BO, we standardize the data to mean zero and a variance of one at each iteration. We further transform the input space specified by the problem intro the hypercube $\mathcal{X} \in [0, 1]^d$. For completeness, we again list all the baselines below and then discuss the optimization of the acquisition functions.

**GPs:** As kernel, we choose a Matérn kernel with $\nu = 2.5$ and use individual lengthscales for all input dimensions that are optimized at every iteration by minimizing the log marginal likelihood [41]. For all lengthscales $\ell_i$ we use box constraints as $\ell_i \in [0.005, 4]$ [28].

**I-BNNs:** Infinite-width Bayesian neural networks (I-BNNs) [33] have shown promising results in recent work [14]. As in Li et al. [14], we set the depth to 3 and initialize the weight variance to 10 and the bias variance to 1.6. Also as in Li et al. [14] we do not optimize the parameters of the kernel.

**VBLL:** For the VBLLs, we use 3 hidden layers with 128 neurons and ELU activation functions for all experiments. We chose this architecture to closely match the ones of BNN baselines in [14]. For the parameterization of the last layer, we adopt the formulation by Harrison et al. [15]. We relearn the features as well as the variational posterior at each iteration unless specified otherwise.

**Acquisition functions:** We use 10 restarts and 512 raw samples for optimizing the acquisition functions UCB and logEI for all models. For TS we optimize the analytic sample of the VBLLs with § random restart using L-BFGS-B [31] as the optimization method. For TS from the non-parametric models, we use the same heuristic as in [28] and generate $\min\{5000, \max\{2000, 200 \cdot d\}\}$ pseudo-random input points from a Sobol sequence and then sample from the high-dimensional multi-variate normal. The next location is then the argmax of the sample path.

# B  Training and Optimization

For training the VBLL models, we closely follow Harrison et al. [15]. For all experiments, we use AdamW [42] as our optimizer with a learning rate of $1e^{-3}$, set the weight decay for the backbone (*not* including the parameters of the VBLL) to $1e^{-4}$, and use norm-based gradient clipping with a value of 1. For the VBLL, we set the prior scale to 1 and the the Wishart scale to 0.01. A sensitivity analysis of the Wishart scale on the performance and run time is in Appendix E. As mentioned in Sec. 2, we employ early stopping for all VBLL models based on the training loss. We track the average loss of a training epoch and if this average loss does not improve for a 100 epochs in a row, we stop training and use the model parameters that yielded the lowest training loss.

# C  On Neural Network Thompson Sampling

We further investigated the use of using neural network based Thompson samples. With VBLLs, we effectively maintain a distribution over plausible deterministic NNs that are in accordance with the current data set and noise level. Instead of maintaining such a distribution and then sampling from the variational posterior of the weights $\boldsymbol{w}$, a straight-forward idea would be to *directly* train a deterministic neural network using the same optimizer (including the same early stopping etc.), and L2 loss, as well as L2 regularization for the backbone directly generating a MAP Thompson sample. Note that with such an approach, one would no longer be able to leverage acquisition functions that rely on good uncertainty quantification such as logEI or more sophisticated such as information-theoretic acquisition functions based on entropy search such as MES [43], which may require large ensembles (up to 100 NNs), making it computationally expensive. Still, we tested this baseline and the results are summarized in Fig. 4.

Here MAP refers to the MAP Thompson sample baseline. We can observe that this simple baseline performs surprisingly well but cannot match the performance of the other baselines. We also observe that in relatively simple problems, such as Branin and Ackley2D, the variance is significantly larger compared to the other baselines, which is undesirable in BO applications. We also compared this baseline on the high-dimensional and real-world benchmarks and the results are shown in Fig. 5.

The MAP approach demonstrates superior performance on the NNdraw task. This is likely because the Wishart scale in the VBLL baselines is not well-tuned for the NNdraw problem (cf. Sec. ). The data is

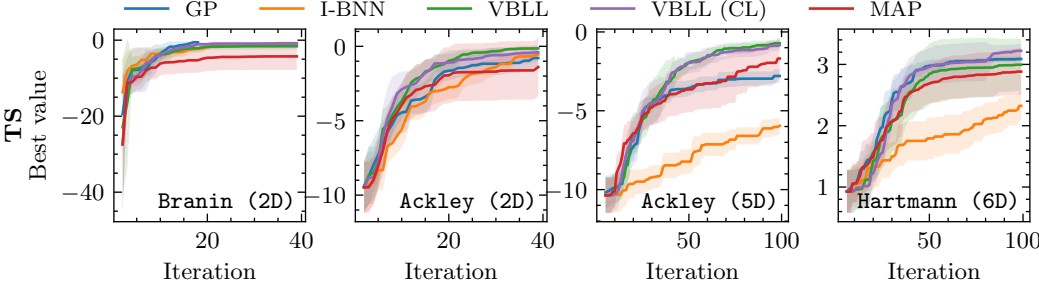

Figure 4: *Comparison of neural network Thompson sampling on synthetic benchmarks.* The deterministic MAP Thompson sample shows surprisingly good performance however also yields large variance on simple benchmarks which is undesirable.

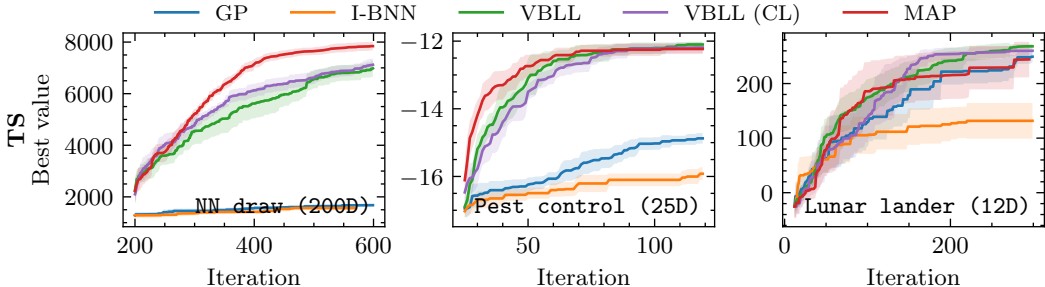

Figure 5: *Comparison of neural network Thompson sampling on high-dimensional and real-world benchmarks.* On these benchmarks, the MAP Thompson sample baseline shows mixed performance. It performs better on `NNdraw` but exibits large variance for `Lunarlander`.

normalized to zero mean and a standard deviation of one; however, with a Wishart scale greater than zero, the assumed noise in this normalized space affects the accuracy of the correlations between data points–especially for the large value range in `NNdraw`. In contrast, the MAP baseline, by design, does not account for noise, which may contribute to its better performance in this context. Additionally, the MAP baseline exhibits slightly faster convergence on `Pestcontrol`. For `Lunarlander`, the MAP approach shows considerable variance and fails to match the final performance of the VBLLs.

## D Experiment Details and Full Results

In the following, we will list details about all the conducted experiments and give the full results for all acquisition functions, i.e., `UCB`, `logEI`, and `TS`.

**Branin:** A standard two dimensional optimization benchmark with three global optima.

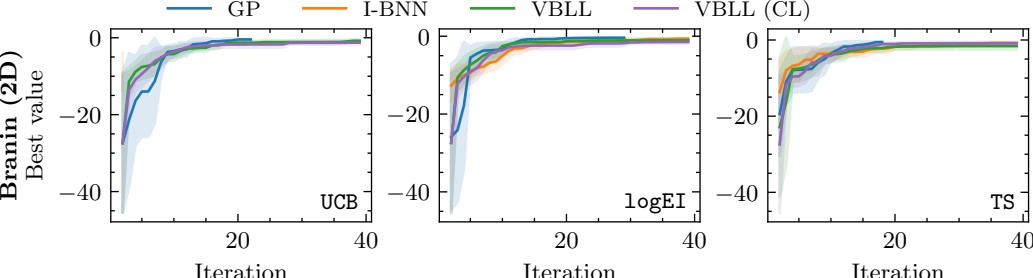

**Ackley:** A standard optimization benchmark with various local optima (depending on the dimensionality) and one global optimum. In out experiments, we compare the surrogates on a 2D and 5D version and set the feasible set to the hypercube $\mathcal{X} = [-5, 10]^d$ as in [28].

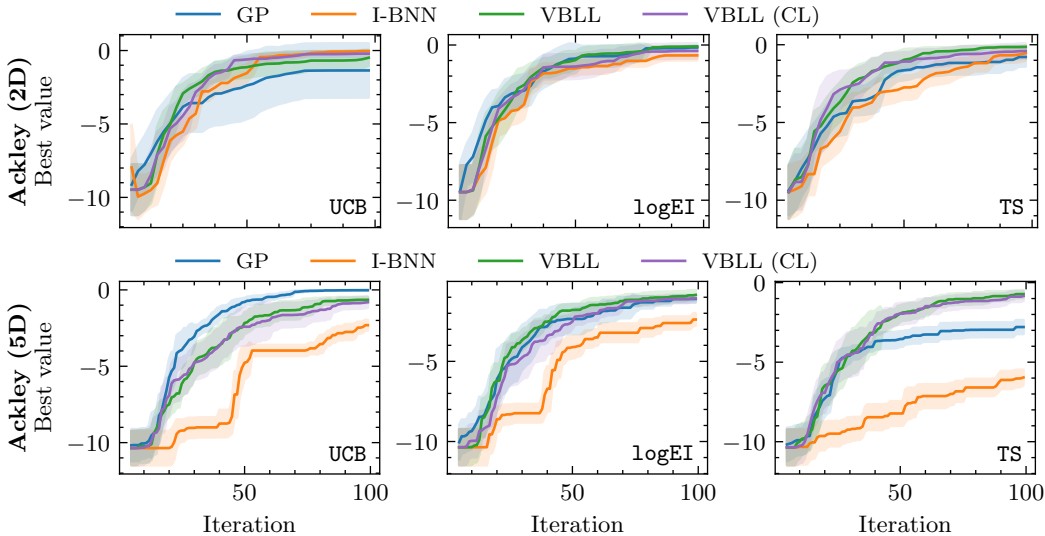

**Hartmann:** A standard six dimensional benchmark with six local optima and one global optimum.

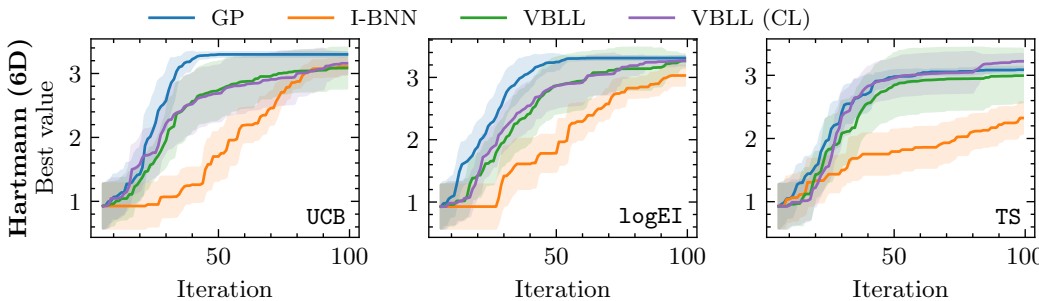

**NN draw:** In this optimization problem, our goal is to find the global optimum of a function defined by a sample from a neural network within the hypercube $\mathcal{X} = [0, 1]^d$. This benchmark was also employed in [14]. We use a fully connected neural network with two hidden layers, each containing 50 nodes, and ReLU activation functions. The input size corresponds to the dimensionality of the optimization problem (in our case, 200), and the output size is one. To generate a function, we sample all weights from the standard normal distribution $\mathcal{N}(0, 1)$. For a fair comparison, we use the same fixed seed across all baselines ensuring that the same objective function is used.

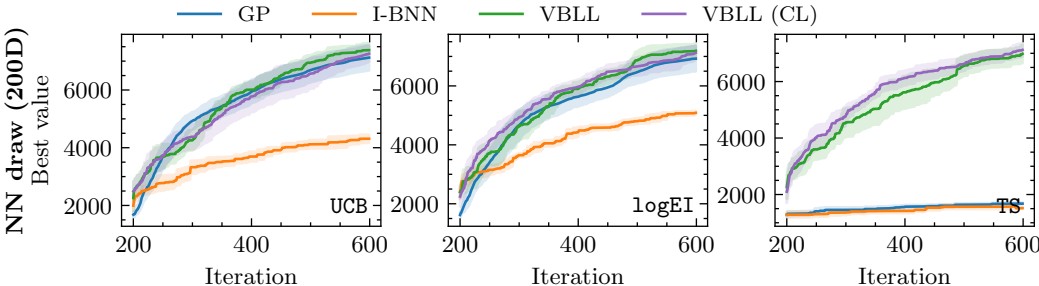

**Pest control:** This optimization problem was also in [14] and aims to minimizing the spread of pests while minimizing the prevention costs of treatment and was introduced in [38]. In this experiment, we define the setting as a categorical optimization problem with 25 categorical variables corresponding to stages of intervention, with 5 different values at each stage. As mentioned in [38], dynamics behind this problem are highly complex resulting in involved correlations between the inputs.

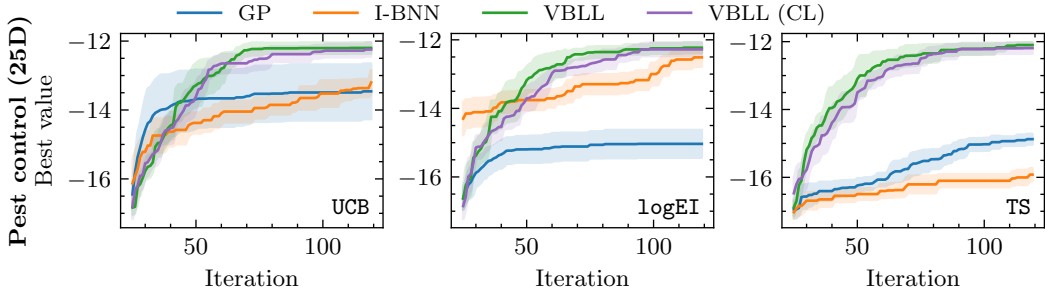

**Lunar lander:** Lunar lander is an environment from OpenAI gym. The objective is to maximize the average final reward over 50 randomly generated environments. For this, 12 continuous parameters of a controller have to be tuned as in [28].

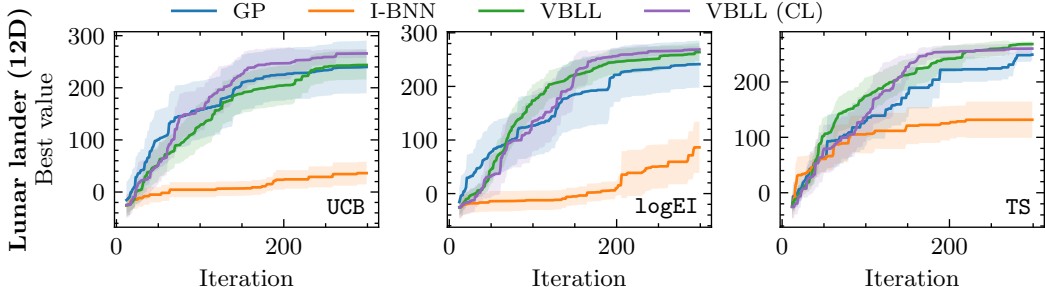

# E   Hyperparameter Sensitivity

The parametric VBLL surrogate has some hyperparameters such as the Wishart scale or the width of the neural network backbone that have to be specified a-priori. In the following, we present results on the hyperparameter sensitivity of the VBLL surrogate model and demonstrate that tuning hyperparameters can improve empirical performance but also that the VBLL surrogate model is rather robust for a wide range of specifications. In Sec. E.1, we will first consider the sensitivity with respect to the Wishart scale and the reinitialization rate for continual learning. Following this, Sec. E.2 then studies the sensitivity regarding the width of the neural network backbone. Lastly, Sec. E.3 considers the robustness to different noise levels.

## E.1   Wishart Scale and Continual Learning Sensitivity

We sweep a number of hyperparameters in the VBLLs in order to experiment with the hyperparameter sensitivity of the VBLL models. In particular we sweep the Wishart Scale and the re-initialization rate of the model. The re-initialization rate determines how often the VBLL model is re-initialized rather than using CL on the backbone and the variational posterior.

The results of the hyperparameter sweep of the Wishart scale and reinitialization rate on Ackley (5D) are in Fig. 6 and on Pestcontrol in Fig. 7. Please note that we have computed running averages on these figures to make qualitative assessment easier. We find that both of these hyperparameters have impact on BO performance, but that the VBLL models are not exceedingly brittle to the values of these hyperparameters. These results also indicate that continual learning for VBLLs is an area of

interest, not only to reduce fitting time, but also because there *appears* to be benefits to not always re-initializing the neural network (see e.g., Fig. 6 (b) for reinitialization rates 3 and 5). Based on these results we also hypothesise that tuning the Wishart scale appropriately for the problem at hand may lead to increased model performance.

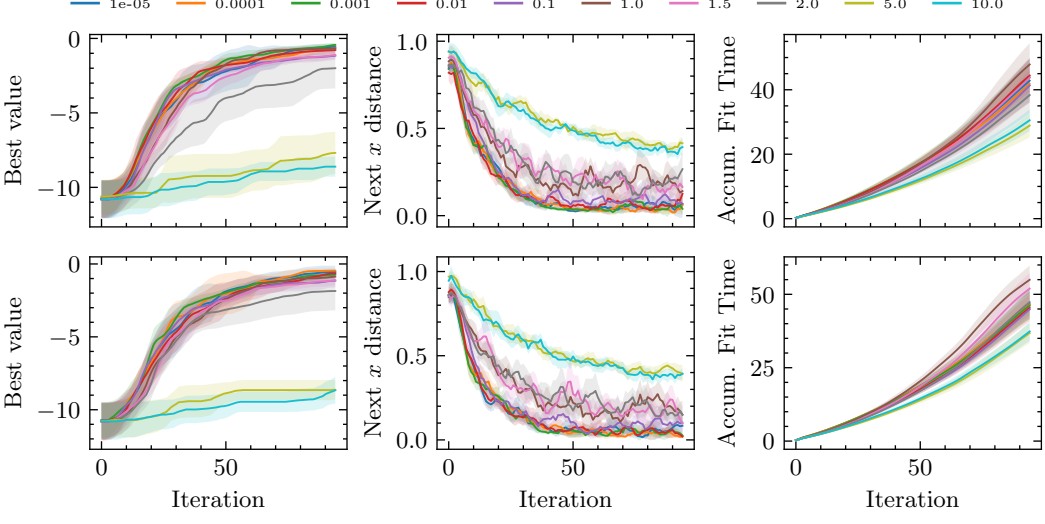

(a) Sensitivity to the Wishart scale (noise free (top), noisy (bottom))

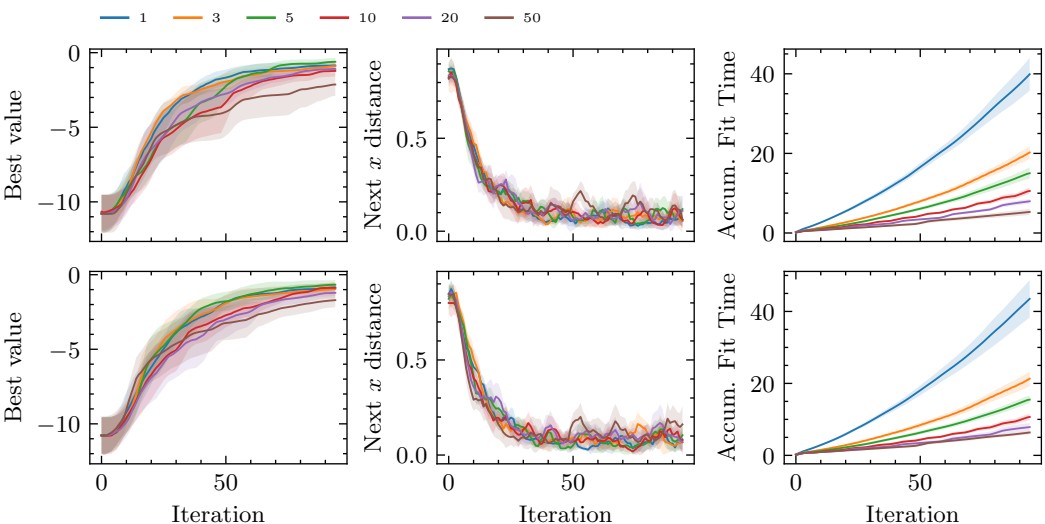

(b) Sensitivity to network reinitialization rate (noise free (top), noisy (bottom))

Figure 6: Hyperparameter sensitivity on the `Ackley5D` benchmark.

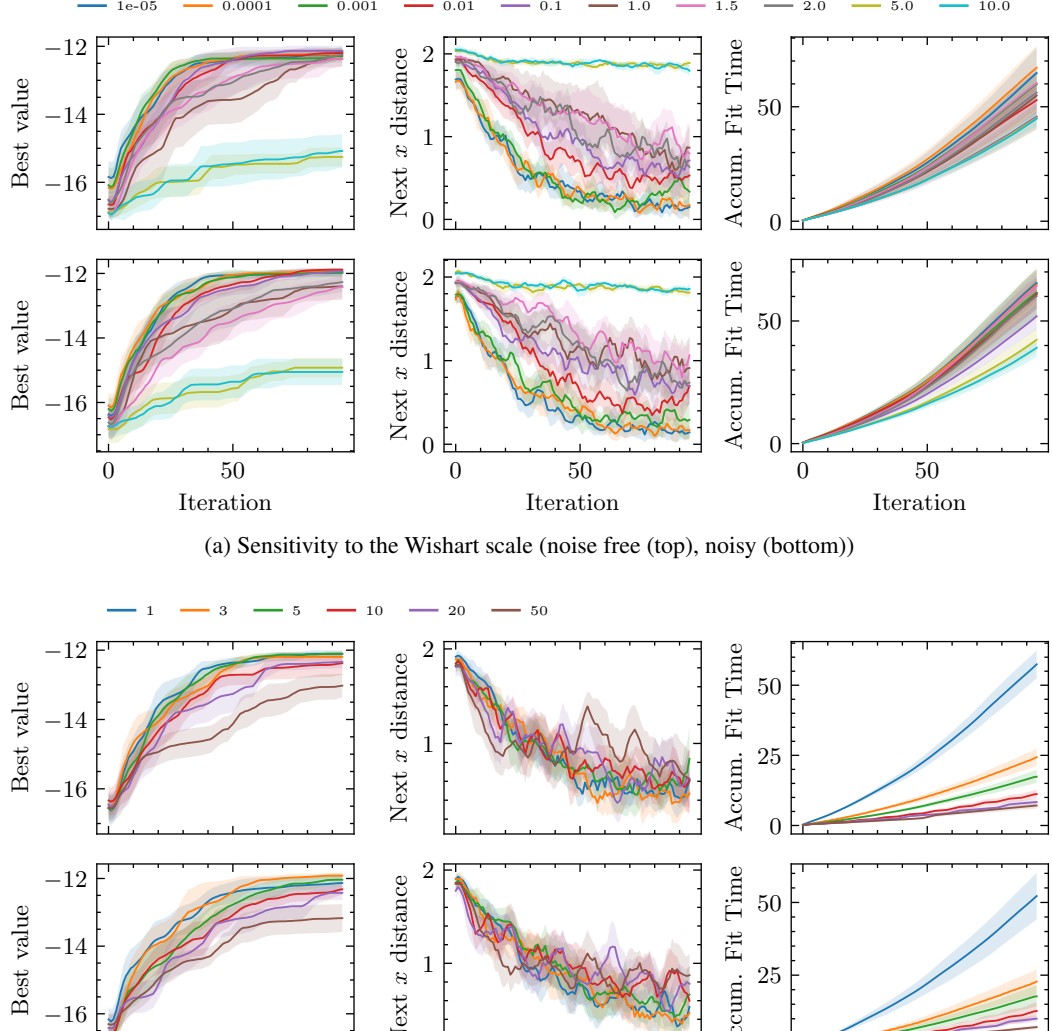

(a) Sensitivity to the Wishart scale (noise free (top), noisy (bottom))

(b) Sensitivity to network reinitialization rate (noise free (top), noisy (bottom))

Figure 7: Hyperparameter sensitivity on the `Pestcontrol` benchmark.

## E.2 Model Width Ablation

To evaluate the impact of model width on the performance of both the MAP and VBLL Thompson sampling methods, we conducted a series of experiments on the `Ackley5D` and `Pestcontrol` benchmarks varying the model width.

As illustrated in Fig. 8 (a) and 9 (a), increasing the model capacity (width) of the MAP baseline results in a significant increase in variance, especially pronounced in the `Ackley5D` benchmark. This high variance suggests that the MAP method is highly sensitive to changes in model width, making it challenging to tune effectively for consistent performance across different tasks.

In comparison, the VBLL method exhibits more robustness to model capacity, as shown in Fig. 8 (b) and 9 (b). Despite increasing the model width, VBLL does not suffer from the high variance observed in the MAP baseline. This robustness is advantageous in practical application where extensive model tuning is unfeasible and hints that VBLL may also perform better when scaling to larger model sizes.

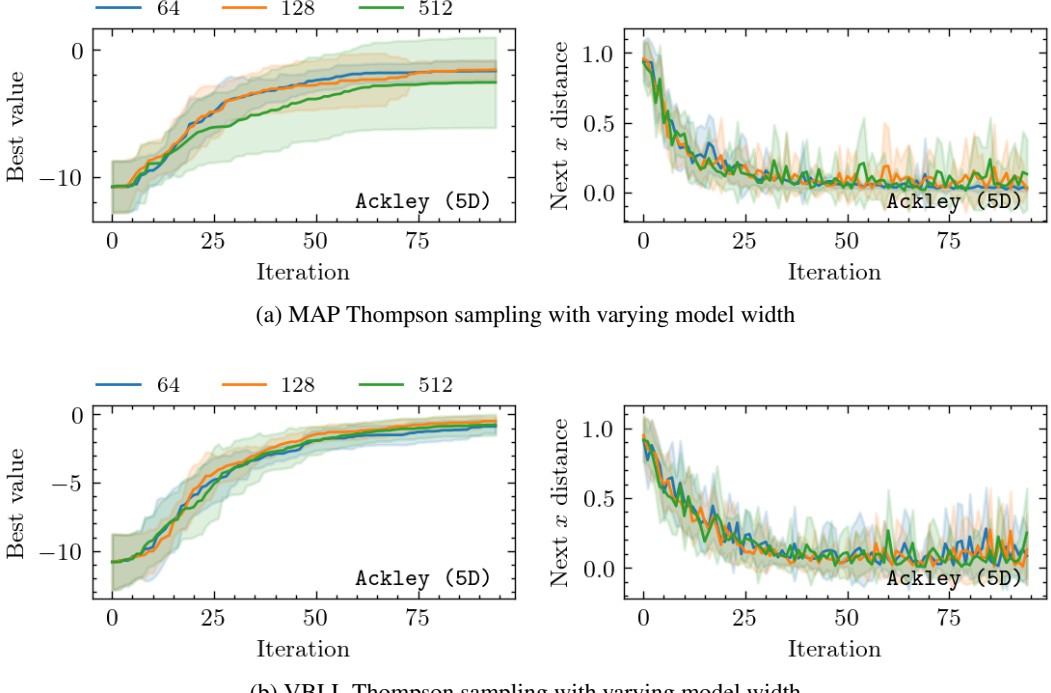

(a) MAP Thompson sampling with varying model width

(b) VBLL Thompson sampling with varying model width

Figure 8: Comparison of neural network Thompson sampling methods on the `Ackley5D` benchmark with varying model width. The models were trained with width of 64, 128 and 512 neurons.

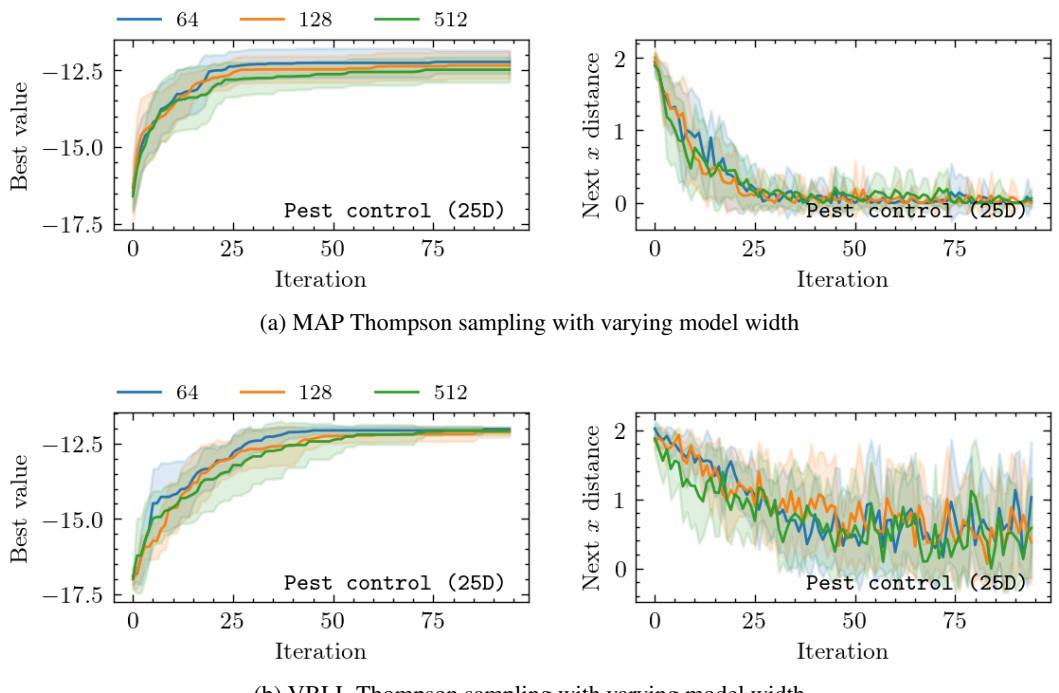

(a) MAP Thompson sampling with varying model width

(b) VBLL Thompson sampling with varying model width

Figure 9: Comparison of neural network Thompson sampling methods on the `Pestcontrol` benchmark with varying model width. The models were trained with width of 64, 128 and 512 neurons.

### E.3 Model Performance in the Presence of Noise

Lastly, we also benchmark the different surrogates on different noise levels. We again only consider `Ackley5D` (Fig. 10) and `Pestcontrol` (Fig. 11). For these experiments, we use the same Wishart scale of $0.01$ for the VBLL baseline. We can observe that all models, besides the MAP baseline in `Ackley5D`, are rather robust the change in noise level.

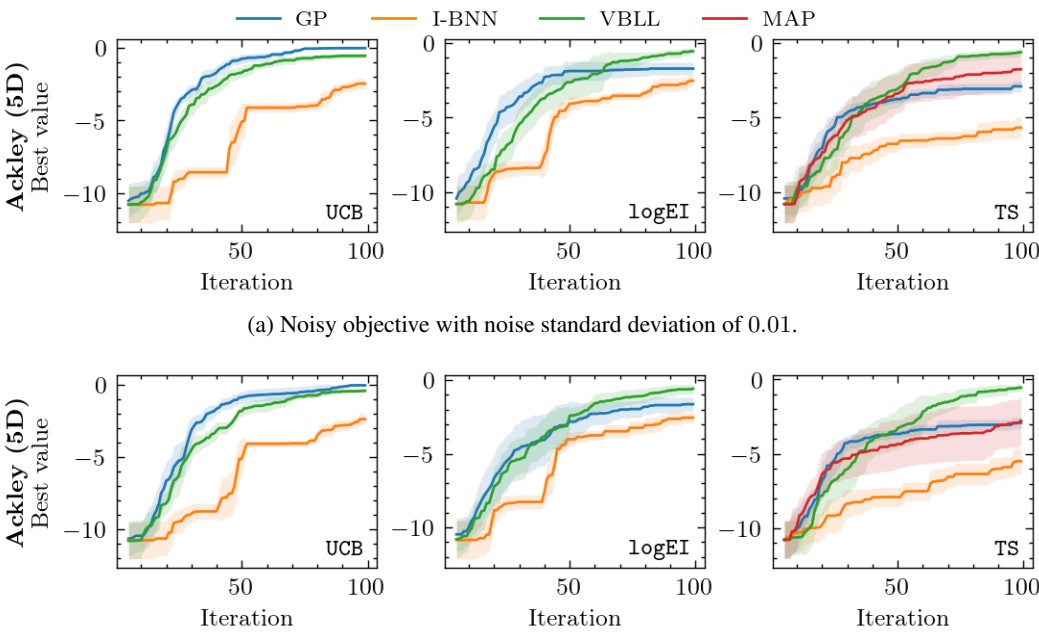

(a) Noisy objective with noise standard deviation of $0.01$.

(b) Noisy objective with noise standard deviation of $0.1$.

Figure 10: Performance comparison of baseline methods on `Ackley5D` benchmark with noise.

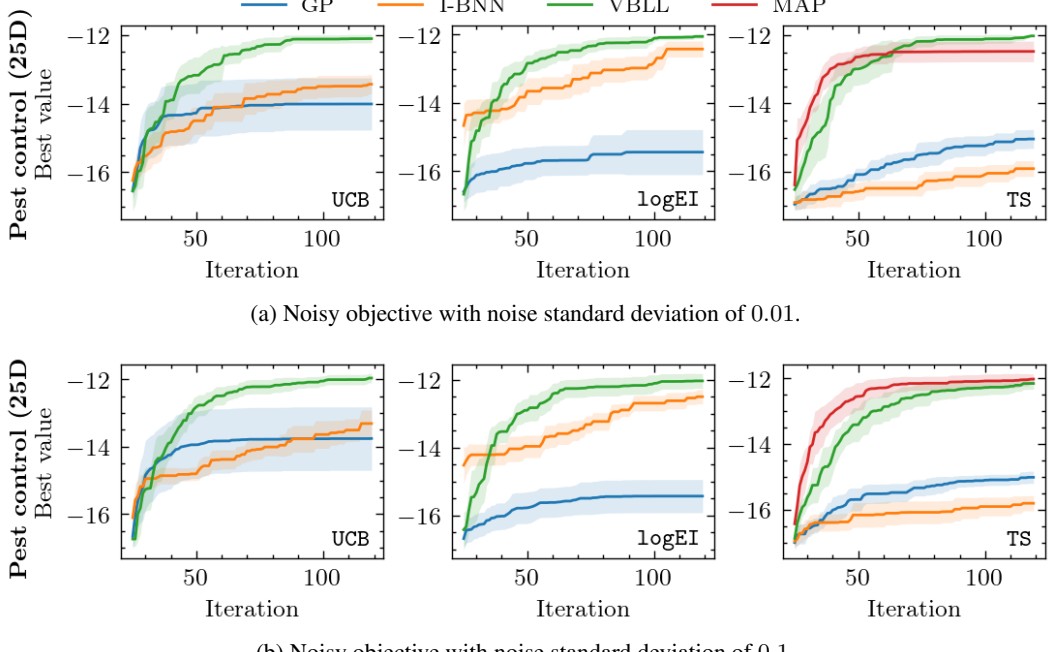

(a) Noisy objective with noise standard deviation of $0.01$.

(b) Noisy objective with noise standard deviation of $0.1$.

Figure 11: Performance comparison of baseline methods on `Pestcontrol` benchmark with noise.

