# OpenReview forum: "Variational Last Layers for Bayesian Optimization"
_NeurIPS.cc/2024/Workshop/BDU — NeurIPS BDU Workshop 2024 Poster_

### Official Review · Reviewer_D7DR · 2024-09-24
**Interesting work using BNNs for BO**

**Rating:** 8
**Confidence:** 4

**Review:**

# Summary

This work proposes a new BNN surrogate for Bayesian optimization, focusing on existing last-layer variational approximation. In practice, the network is treated as a nonlinear feature extractor which then become the covariates for a generalized linear model (GLM) which under Gaussian priors for the linear weights can be solved analytically, yielding Gaussian predictions. Thus, standard acquisition functions can be applied. Despite being a workshop paper, the submission is quite thorough in testing the method against various baselines (GPs and other BNNs) on a variety of test functions, both synthetic and more challenging high-dimensional problems.

# Comments

This is a great workshop paper. The basic idea is fairly simple - use (variational) last-layer BNNs as a surrogate model for good old Bayesian optimization (nothing fancy). This is straightforward and perhaps not *particularly* original, but still novel to my knowledge. Still, the execution is very good, with high clarity and interesting results that will be significant for the community. The quality is very high with a broad set of experimental tests, both in terms of functions and many reasonable baselines and ablation studies, with a comprehensive appendix.

The authors perhaps could focus more on the limitations of the work. For example, the cost or retraining the network seems larger than updating a GP, at least when the number of data point is small; although the authors do suggest using warm-starting which they call "continual learning". Also, the paper mentions that optimization of the sampled network for Thompson sampling is "analytical" but unclear what they mean here (I assume they mean they can run a first-order optimizer directly on the sampled surrogate, which is not immediately feasible for a GP sampled path; clearly there is no closed-form solution to the optimization problem).

Overall, a great workshop contribution and very appropriate for the themes of the workshop.

---

### Official Review · Reviewer_PJSr · 2024-09-28
**Review of Submission 83**

**Rating:** 4
**Confidence:** 3

**Review:**

**Summary:** This work introduces a novel probabilistic model for Bayesian optimization based on neural networks with variational Bayesian layers. The model consists of a neural network for embedding and a final layer represented by a traditional Bayesian linear regression model. While posterior inference could be achieved by optimizing the analytically tractable log-likelihood, the authors propose optimizing a recently introduced variational lower bound instead, enhancing computational efficiency. Numerical experiments across various tasks demonstrate that the proposed approach performs competitively against traditional Gaussian processes and a recent neural network-based alternative.

**Strengths:** The paper is well-written and the proposed method is technically sound.

**Weaknesses:** The novelty of the work appears limited. If my understanding is correct, the only difference between this approach and that of Snoek et al. (2015) is the use of the variational lower bound optimization instead of direct log-likelihood optimization. Furthermore, the lack of a comparison with this former method makes it difficult to assess the benefits of the proposed modification.

---

### Decision · Program_Chairs · 2024-10-09

Accept (Poster)